# Preparation of the Carbonized Zif−8@PAN Nanofiber Membrane for Cadmium Ion Adsorption

**DOI:** 10.3390/polym14132523

**Published:** 2022-06-21

**Authors:** Hui Sun, Jiangli Feng, Yaoyao Song, Lei Xu, Xiaogang Cui, Bin Yu

**Affiliations:** 1College of Textiles Science and Engineering, Zhejiang Sci-Tech University, 928 Second Avenue, Xiasha Higher Education Zone, Hangzhou 310018, China; sunhui@zstu.edu.cn (H.S.); 18758094708@163.com (J.F.); sunny@soyang.net (Y.S.); 1981_xl@163.com (L.X.); c503950252@163.com (X.C.); 2School of Textile and Clothing and Art and Media, Suzhou Institute of Trade & Commerce, 287 Xuefu Road, Suzhou 215009, China

**Keywords:** electrospun nanofiber membrane, ZIF−8, carbonization, cadmium ion, adsorption

## Abstract

The zeolitic imidazolate framework (ZIF−8)@polyacrylonitrile (PAN) nanofiber membrane was prepared and carbonized for heavy metal cadmium ion (Cd^2+^) adsorption in aqueous medium. Zinc oxide (ZnO) was first sputtered onto the surface of the PAN electrospun nanofiber membrane to provide a metal ion source. Then, the ZIF−8@PAN nanofiber membrane was prepared via in situ solvothermal reaction and carbonized in a tube furnace at 900 °C under a N_2_ atmosphere to enhance adsorption performance. The synthesized ZIF−8 particles with polyhedral structure were uniformly immobilized on the surface of the PAN electrospun nanofiber membrane. After being heated at 900 °C, the polygonal ZIF−8 shrank, and the carbonized ZIF−8@PAN nanofiber membrane was obtained. Compared with the nanofiber membrane without being carbonized, the adsorption capacity of the carbonized ZIF−8@PAN nanofiber membrane reached 102 mg L^−1^, and its Cd^2+^ adsorption efficiency could be more than 90% under the adsorption temperature of 35 °C and solution of pH = 7.5 conditions. According to the adsorption thermodynamics analysis, the Cd^2+^ adsorption process of the carbonized ZIF−8@PAN nanofiber membrane was spontaneous. The whole Cd^2+^ adsorption process was more suitably described by the pseudo second-order adsorption kinetics model, indicating that there exists a chemical adsorption mechanism besides physical adsorption.

## 1. Introduction

With the development of the global economy and continuous progress of light industry, the discharge of industrial waste water has rapidly increased. The pollutants in waste water containing heavy metal ions have entered and accumulated in drinking water resources and the soil [1,2], which not only affects the output of agricultural products, but also ultimately endangers human health through the food chain [3,4,5]. The existing methods used to remove heavy metal pollutions mainly include filtration [6], adsorption [7,8], membrane separation [9], ion exchange [10] and photocatalytic degradation [11]. Among them, the adsorption separation has been one of the most popular methods on account of its effectiveness and simplicity.

Nanofiber membranes produced by the electrospinning technique are of high specific surface area and porosity due to their low diameter of several tens of nanometers to few hundred nanometers. As a result, they have been widely used in the field of adsorption and separation, air filtration, drug delivery system, tissue engineering, antimicrobial applications, catalysis, energy storage devices, and controllable water absorption [12,13,14,15,16,17,18,19]. Polyacrylonitrile (PAN) is a kind of polymer material with abundant functional cyano−groups (–CN) on its macromolecular chain, which results in its excellent thermal and chemical stability, and resistance to solvent, oxidative degradation, sunlight, and weather. Additionally, it can be used as precursor for carbon fibers [20]. Electrospinning of PAN into nanofiber membrane may expand its applications in adsorption and separation due to the obtained nano−structural characteristics. However, pure PAN nanofiber membrane do not have sufficient adsorption ability of heavy metal ions [21,22]. Functional modification is a good choice to improve the adsorption performance of PAN nanofiber membrane [22,23].

Over the past few decades, metal–organic frameworks (MOFs) have attracted tremendous interest as a new class of porous materials because of their advantages of high porosity, low density, large specific surface area, regular channel, adjustable pore size, and structural diversity [24,25]. They are composited by metal ion sources and organic ligands, where metal ions are connection sites, and organic ligands support the three−dimensional extension of space. Their controllable structures and designable components could guarantee a high adsorption ability with their abundant active sites, which allows MOFs to be used as one of the suitable candidates in effective heavy metal ion adsorption [24,25,26]. Therefore, MOFs are ideal materials for functional modification of PAN to obtain efficient heavy metal adsorption ability. Efome et al. [27] prepared an electrospun nanofiber composite membrane for the adsorption of Cd^2+^and Zn^2+^ in water via the co−electrospinning of PAN and water−stable Zr−based MOF−808. They found that the maximum adsorption capacities of PAN/MOF−808 with pore size of 0.5 to 1 μm for Cd^2+^ and Zn^2+^ were 225.05 and 287.06 mg g^−1^, respectively. The high separation performance and reusability of the PAN/MOF−808 membrane and the outstanding water stability of MOF−808 suggested that this membrane has potential application prospects for water treatment. Jamshidifard et al. [28] synthesized UiO−66−NH_2_ MOF by the microwave heating method and incorporated this MOF into the PAN/chitosan nanofiber membrane for the removal of Pb^2+^, Cd^2+^, and Cr^6+^ from aqueous solutions through adsorption and membrane filtration processes. It was found that the prepared PAN/chitosan/UiO−66−NH_2_ nanofibrous adsorbent had an average fiber diameter of 235 nm. The maximum monolayer adsorption quantities of the adsorbent for Pb^2+^, Cd^2+^, and Cr^6+^ ion sorption were 441.2, 415.6, and 372.6 mg g^−1^ under the optimal condition, respectively. When the simultaneous adsorption was carried out, the metal ion sorption was in order of Pb^2+^ > Cd^2+^ > Cr^6+^. Yang et al. [29] fabricated a zeolitic imidazolate framework−8 (ZIF−8)@ZIF−8/PAN nanofiber membrane with a high surface area of 871.0 m^2^ g^−1^ by the incorporation of ZIF−8 with PAN electrospun nanofiber membrane, followed by the in situ growth of ZIF−8 onto the surface of the nanofibers for effective adsorption and reduction to Cr^6+^ from aqueous media. The results showed that Cr^6+^ could be effectively removed with the maximum adsorption capacity of 39.68 mg g^−1^ and with better recyclability. Moreover, toxic Cr^6+^ could be further reduced partially to the relatively harmless Cr^3+^ on the basis of the proposed feasible adsorption–reduction mechanism. Peng et al. [30] mixed zinc acetate, dimethylimidazole, and polyethylene to form the ZIF−8 precursor, and they immobilized this precursor on the surface of a PAN nanofiber membrane by hot pressing to fabricate ZIF−8@PAN adsorbent for Cu^2+^ removal. The average pore size of the ZIF−8@PAN adsorbent was 2.66 nm, and its mass loss was approximately 40.0% at 280 °C. The fabricated adsorbent showed good performances with fast flux (12,000 L/(m(^2^)h)) and high filtration efficiency (96.5%) for Cu^2+^. Zhang et al. [31] first prepared a UiO−66−(COOH)_2_/PAN nanofibrous substrate by electrospinning and then cast a calcium alginate layer onto the nanofibrous substrate surface by using an automated coating applicator for the removal of Pb^2+^ in the waste water. They found that with the increase of MOF doping, the surface roughness of PAN−UiO−66−(COOH)_2_ nanofiber was enhanced, and more and more MOF particles were wrapped tightly in nanofibers. The Pb^2+^ adsorption capacity of the resulting composite membrane was 195.9 mg g^−1^.

ZIF−8 is a subclass of MOFs made of metal zinc ions (Zn^2+^) tetrahedrally coordinated to 2−methylimidazole organic linker. It is of great promise for removing contaminants from water in consideration of its excellent water and thermal/chemical stability [32,33]. Especially after being carbonized at high temperatures, ZIF−8 can partly generate nano−porous carbon, which further increases specific surface area and adsorption capacity [34,35]. However, in the traditionally solvothermal synthesis reaction of ZIF−8, the metal ion source is provided by some explosive or toxic chemicals, for example, zinc nitrate hexahydrate (Zn (NO_3_) _2_·6H_2_O) [27,28,29] or zinc acetate [30]. Herein, our paper considers an effective and environmentally friendly way to provide Zn^2+^ as a metal ion source for the synthesis of ZIF−8.

In this paper, magnetron sputtering technology was used to sputter a uniform zinc oxide (ZnO) coating on the PAN electrospun nanofiber membrane as a Zn^2+^ source required for the subsequent ZIF−8 synthesis reaction. Then, 2−methylimidazole was used as a ligand, and ZIF−8 was immobilized on the surface of PAN electrospun nanofiber membrane through the in situ solvothermal synthesis method. Herein, dangerous chemicals were avoided, and the fabrication of ZIF−8@PAN composite nanofiber membrane became safer and simpler. Finally, the carbonized ZIF−8@PAN nanofiber membrane was prepared by high−temperature calcination to improve the adsorption performance of Cd^2+^ in aqueous medium.

## 2. Materials and Methods

### 2.1. Materials

N,N−dimethylformamide (DMF) and ethanol were purchased from Hangzhou Gaojing Fine Chemical Industry Co., Ltd. (Hangzhou, China). ZnO target (purity 99.99%) was obtained from Beijing Yijie Material Technology Co., Ltd. (Beijing, China). 2−methylimidazole (98%) was purchased from Aladdin Industrial Corporation (Shanghai, China). Cadmium chloride was bought from Shanghai Macklin Biochemical Co., Ltd. (Shanghai, China). PAN with a weight−average molecular weight of 150,000 was provided from Hangzhou Acrylon Co., Ltd. (Hangzhou, China). Deionized water and PAN electrospun nanofiber membrane with a thickness of about 200 μm were prepared in our laboratory. All chemicals were used as received without further purification.

### 2.2. Preparation of ZnO@PAN Nanofiber Membrane

Prior to electrospinning, PAN was dissolved in DMF to obtain a spinning dope with a concentration of 12%. Then, the prepared spinning dope was put into a 10 mL disposable syringe and electrospun. The electrospinning flow rate was 0.05 mL/min. The needle tip–collector distance was 12 cm, and the electrospinning voltage was 12 kV. During the electrospinning, the solvent was volatilized at room temperature. After that, a uniform ZnO coating was initially sputtered on the obtained PAN electrospun nanofiber membrane using magnetron sputtering technology. The PAN electrospun nanofiber membrane with a diameter of 4 cm was scissored and fixed on the disc, while the ZnO target was fixed on the target cavity of the instrument. The parameters of the radio frequency (RF) magnetron sputtering model were set at room temperature. The vacuum degree was 1.0 × 10^−4^ Pa, and the pressure was kept at 1.33 Pa. The sputtering power was 100 W, and the sputtering time was 40 min. Then, the ZnO@PAN nanofiber membrane was obtained for the next step.

### 2.3. Preparation of ZIF−8@PAN Nanofiber Membrane

The ZIF−8@PAN nanofiber membrane was fabricated via in situ solvothermal reaction in a high−pressure reactor. A 2 cm × 2 cm ZnO@PAN nanofiber membrane was cut and immersed in a mixed solution composed of DMF and 2−methylimidazole with a concentration of 1%. After being completely immersed, ZnO@PAN nanofiber membrane was placed in the inner cavity of the high−pressure reactor at a temperature of 100 °C for 7 h. After that, the nanofiber membrane was dried at 80 °C for 2 h under vacuum condition, and the ZIF@PAN nanofiber membrane was obtained.

### 2.4. Preparation of the Carbonized ZIF−8@PAN Nanofiber Membrane

The carbonized ZIF−8@PAN nanofiber membrane adsorbent was prepared by the calcination of the resulting ZIF−8@PAN nanofiber membrane at a certain high temperature in a tube furnace. After having been dried, the ZIF−8@PAN composite nanofiber membrane was heated in a tubular calciner at 900 °C for 2 h with a heating ramp rate of 5 °C min^−1^ under a N_2_ flow. The carbonized ZIF−8@PAN nanofiber membrane was finally obtained and is marked by 900 °C−ZIF−8@PAN in the figures. The above synthetic and preparation route are described as follows in Figure 1.

### 2.5. Characterization

X−ray diffraction (XRD, Rigaku D/MAX, Tokyo, Japan) was carried out by using the Cu K_α_ (λ = 1.5406 Å) radiation source in the diffraction angle (2*θ*) range of 5–50° by a step of 5° min^−1^ at 40 kV and 40 mA. Fourier transform infrared spectroscopy (FTIR, Nicolet 57000, Thermo Scientific, Waltham, MA, USA) of the samples were recorded in the spectral range of 4000–400 cm^−1^ using the potassium bromide disk method. Raman spectrum (LabRam ARAMIS, Oxford, UK) was performed in the spectral range of 500–2000 cm^−1^. The excitation wavelength was 514 nm. The morphologies of the samples were observed by field emission scanning electron microscopy (SEM, JSM−5610LV, Tokyo, Japan). The samples were cut into small pieces and fixed on the copper sample stage by use of conductive tape. All of the samples were plated with a thin layer of gold before inspection. The acceleration voltage used was 3 kV. Moreover, energy dispersive X−ray spectrometry (EDS) mode was used to identify the samples.

### 2.6. Adsorption Performance

The Cd^2+^ adsorption performance of the ZIF−8@PAN nanofiber membrane adsorbent before and after carbonization were evaluated by atomic absorption spectrograph (AAS, Varian AA−110, Centennial, CO, USA). Cd^2+^ mother liquor with a concentration of 20 mg L^−1^ was configured and diluted into 10 mg L^−1^ standard solutions with a pH value of 6.5 by hydrochloric acid. Then this Cd^2+^ standard solution with a volume of 450 mL was put into a 500 mL beaker. Moreover, about 40 mg adsorbent was completely immersed in this beaker. After that, this beaker was sealed and put in a 20 °C water bath for about 72 h to ensure that the adsorption had reached equilibrium. Finally, the solution was separated from the adsorbent by means of the high−speed centrifuge with a speed of 5000 rpm, and the residual Cd^2+^ concentration in the supernatant was analyzed by AAS. The intensity (absorbance) of the adsorption peak at 228.8 nm was recorded. The Cd^2+^ adsorption efficiency was calculated by the following equation:(1)$$\mathrm{Adsorption}\;\mathrm{efficiency}\left(\%\right)=\frac{C_{0}-C_{t}}{C_{0}}\times 100\%$$
where *C*_0_ is the initial concentration of Cd^2+^ standard solution; *C_t_* is the Cd^2+^ concentration at a certain adsorption time *t*.

To explore the effect of water bath temperature during the adsorption process on the Cd^2+^ adsorption performance of the carbonized ZIF−8@PAN nanofiber membrane, the above test steps were repeated at various water bath temperatures (20 °C, 25 °C, 35 °C and 45 °C) under pH = 6.5 condition. The adsorption thermodynamics of the carbonized ZIF−8@PAN nanofiber membrane for Cd^2+^ was investigated at three temperatures of 20 °C (293 K), 25 °C (298 K), and 35 °C (308 K), respectively. The Gibbs free energy formula [36,37] was used:(2)$$\mathrm{In}K_{d}=\frac{\Delta S^{\theta }}{R}-\frac{\Delta H^{\theta }}{RT}$$
(3)$$\Delta G^{\theta }=\Delta H^{\theta }-T\Delta S^{\theta }$$

In Equation (2), *T* (K) is the Kelvin temperature, and *R* = 8.314 J mol^−1^ K^−1^ and stands for the universal gas constant. *K_d_* is the thermodynamic equilibrium constant of the adsorption process, which can be calculated from the ratio of *Q_e_* (mg·g^−1^) to *C_e_* (mol·L^−1^). *Q_e_* represents the equilibrium adsorption capacity of Cd^2+^, and *C_e_* is the equilibrium concentration of the solution. *Q_e_* can be calculated by the following formula:(4)$$Q_{e}=\frac{(C_{0}-C_{t})V}{m}$$
where *m* is the mass of the adsorbent, and *V* is the volume of the Cd^2+^ solution. In Equation (3), Δ*H^θ^* and Δ*S^θ^* are the adsorption enthalpy change and adsorption entropy change, respectively, and can be obtained by the plot of ln*K_d_* vs. 1/*T*.

Additionally, the effect of pH value of the Cd^2+^ standard solution on the Cd^2+^ adsorption performance of the carbonized ZIF−8@PAN nanofiber membrane was also discussed under certain adsorption temperature. By use of hydrochloric acid and sodium hydroxide, the pH value of the Cd^2+^ standard solution with a concentration of 10 mg L^−1^ were adjusted to 5.5, 6.5, 7.5, and 8.5 at 20 °C, respectively. Then, the Cd^2+^ adsorption performance of the carbonized ZIF−8@PAN nanofiber membrane were tested. Furthermore, the adsorption mechanism was studied with the pseudo first-order [38,39] and second-order [40] adsorption equations, respectively.

The pseudo first-order adsorption kinetic equation is as follows:(5)$$\mathrm{In}\left(Q_{e}-Q_{t}\right)=\mathrm{In}\left(Q_{e}\right)-k_{1}t$$

The pseudo second-order adsorption equation is as follows:(6)$$\frac{t}{Q_{t}}=\frac{1}{k_{2}Q_{e}^{2}}+\frac{t}{Q_{e}}$$
where *k*_1_ and *k*_2_ are the pseudo first-order and second-order adsorption rate constants, respectively.

## 3. Results and Discussion

### 3.1. Characterization of Adsorbents

Figure 2 shows SEM images of the original PAN, ZIF−8@PAN, and the carbonized ZIF−8@PAN nanofiber membrane, respectively. As seen in Figure 2a, the surface of the PAN nanofibers was relatively smooth, and the average fiber diameter was about 163 nm by use of the Image J software analysis. After the solvothermal reaction, as shown in Figure 2b, a layer of ZIF−8 polyhedral particles with uniform size was attached onto the surface of the PAN electrospun nanofiber membrane. The enlarged image in the upper right corner on Figure 2b clearly shows that the particles had a rhombic dodecahedron structure, which is consistent with the morphology of ZIF−8 reported in the references [1,2]. This phenomenon implies that ZIF−8 might be synthesized on the surface of the PAN nanofiber membrane via in situ growth. After being carbonized at 900 °C, ZIF−8 on the surface of PAN nanofiber membrane still maintained the rhombic dodecahedron structure, as shown in Figure 2c. However, the profile of the polygonal ZIF−8 particles shrank, which may be attributed to the structural damage of ZIF−8 in view of Zn evaporation at high carbonizing temperature [41]. On the other hand, the morphology of PAN fibers changed very little.

The element species of the nanofiber membrane before and after functional modification was performed by EDS to confirm the carbonization of ZIF−8, and the results are shown in Figure 3. In Figure 3a, there were just C and N elements for the original PAN electrospun nanofiber membrane. For the ZIF−8@PAN nanofiber membrane, as seen in Figure 3b, Zn and O elements were introduced, reflecting that the ZIF−8 particles existed on the surface of the PAN nanofiber membrane. After high temperature calcination, the Zn element in ZIF−8 disappeared, as shown in Figure 3c. Hsiao et al. [41] verified that the Zn or ZnO in the ZIF−8 structure can be removed and porous carbon generated after ZIF−8 is calcinated between 800 and 1000 °C. Thus, the result shown in Figure 3c suggests that ZIF−8 on the surface of the PAN nanofiber membrane was carbonized.

To further explore the structure of the carbonized ZIF−8@PAN nanofiber membrane, XRD analysis was performed at room temperature and is plotted in Figure 4. In Figure 4, for the ZIF−8@PAN nanofiber membrane, there were multiple diffraction peaks at around 7.4°, 10.5°, 12.8°, 14.8°, 16.5°, and 18.1°, which were identified with the characteristic peaks of the simulated XRD pattern of ZIF−8 from the CCDC archive (602538). These diffraction peaks were attributed to (011), (002), (112), (022), (013), and (222) crystal planes of ZIF−8, respectively [34,35,42]. Moreover, two strong characteristic peaks could be seen at 38.3° and 44.5°, which were caused by the existence of aluminum foil as the support material for the nanofiber membrane. After the ZIF−8@ PAN nanofiber membrane was carbonized at 900 °C, the corresponding characteristic diffraction sharp peaks of ZIF−8 disappeared, and the diffraction peaks reflecting the amorphous structure appeared. The reason is that the Zn element disappeared on account of the evaporation at high temperature, causing the crystalline structure of ZIF−8 to be destroyed. In the reported research by Abbsi et al. [34], the same phenomenon was also observed. Abbsi et al. found that the characteristic peaks of ZIF−8 disappeared, and the diffraction peaks of the amorphous graphitized structure were produced when ZIF−8 was heated to about 908 °C, meaning that the Zn element was evaporated, and ZIF−8 was carbonized. Thus, it may be speculated that our ZIF−8@PAN nanofiber membrane was carbonized at 900 °C. The carbonized ZIF−8 has a higher surface area, which is helpful to promote absorption performance [34,35,41].

FTIR spectra of the ZIF−8@PAN nanofiber membrane before and after carbonization are shown in Figure 5a. The FTIR spectrum of the ZIF−8@PAN nanofiber membrane revealed the characteristic absorption peaks at 1145 cm^−1^ and 995 cm^−1^, which is attributed to C–N bonds. The absorption peak appearing at 1585 cm^−1^ was due to the stretching vibration of the C=N bond on the benzene ring. Additionally, the two absorption peaks around 665 cm^−1^ and 743 cm^−1^ were attributed to the stretching vibrations of =CH and N–H, respectively. These results further illustrate that the ZIF−8@PAN nanofiber membrane was obtained [43,44]. For the carbonized ZIF−8@PAN nanofiber membrane, the relevant characteristic peaks of ZIF−8 disappeared. Moreover, two new characteristic peaks appeared at 1560 cm^−1^ and 1260 cm^−1^, which is explained by the formation of a porous carbon structure after high temperature calcination of the ZIF−8@PAN nanofiber membrane [45].

The Raman spectrum of the carbonized ZIF−8@PAN nanofiber membrane was carried out and is plotted in Figure 5b. As shown in Figure 5b, both the G peak at 1584 cm^−1^ and the D peak at 1330 cm^−1^ can be clearly seen. Generally, the G peak is caused by the stretching motion of a carbon atom with sp^2^ hybridization in carbon rings or a carbon chain. The D peak is induced by the disorder and defect structures of the materials. The appearance of the G peak and D peak confirmed that the structure of ZIF−8 on the surface of PAN became unperfect after being carbonized at 900 °C, which matches with that of XRD analysis.

### 3.2. Cd^2+^ Adsorption Performance

The Cd^2+^ adsorption performance of the ZIF−8@PAN nanofiber membrane before and after carbonization were evaluated and compared with the original PAN electrospun nanofiber membrane, as shown in Figure 6. It can be seen in Figure 6 a,b that when the original PAN nanofiber membrane was used as an adsorbent, the Cd^2+^ concentration in the solution was hardly changed, and its *Q_e_* value was zero, indicating that the original PAN nanofiber membrane could not adsorb Cd^2+^. The ZIF−8@PAN nanofiber membrane exhibited limited Cd^2+^ adsorption ability, and its *Q_e_* value and adsorption efficiency were about 25 mg L^−1^ and 20% after adsorption equilibrium was reached, respectively, as shown in Figure 6b,c. For the carbonized ZIF−8@PAN nanofiber membrane, as the adsorption time increased, the Cd^2+^ concentration quickly declined in the initial eight hours. Thereafter, the adsorption rate slowed down until about 50 h, as shown in Figure 6a. It can be observed in Figure 6b that when the Cd^2+^adsorption equilibrium was obtained, the *Q_e_* value increased to 99 mg L^−1^, and the adsorption efficiency could reach about 88%. These phenomena manifest that compared with the ZIF−8@PAN nanofiber membrane, the carbonized ZIF−8@PAN nanofiber membrane had a higher Cd^2+^ adsorption ability, which is attributed to the increased contact area between the adsorbent and Cd^2+^ after the ZIF−8@PAN nanofiber membrane was carbonized.

### 3.3. Effect of Temperature

Temperature plays an important role in the whole adsorption process. Thus, the effect of adsorption temperature on the Cd^2+^ adsorption performance of the carbonized ZIF−8@PAN nanofiber membrane was discussed and is plotted in Figure 7. Under the condition of pH = 6.5, when the Cd^2+^adsorption system was put into a water bath with different temperatures, both the *Q_e_* value and the adsorption efficiency of Cd^2+^ first increased with the increasing of the water bath temperature from 20 °C to 35 °C. At the adsorption temperature of 35 °C, the *Q_e_* value and the adsorption efficiency were about 89 mg g^−1^ and 79%, respectively. However, when the adsorption temperature was more than 35 °C, the *Q_e_* value and adsorption efficiency reduced. In fact, during the adsorption process, there existed a competing relationship between adsorption and desorption. In the near room temperature range, with the increasing of the adsorption temperature, the intensity of Cd^2+^ Brownian movement increased while the Cd^2+^ solution viscosity decreased, meaning the more contact opportunity between Cd^2+^ and the surface of the carbonized ZIF−8@PAN nanofiber membrane. At this time, the Cd^2+^ adsorption rate was faster than its desorption rate. However, the Cd^2+^ desorption effect was dominant at the adsorption temperature above 35 °C. The reason is that the higher temperature may have led to the more violent movement of Cd^2+^ and provided enough energy to the Cd^2+^ desorption from the adsorbent. Thus, it is believed that the Cd^2+^ adsorption effect is best when the adsorption temperature is 35 °C.

### 3.4. Adsorption Thermodynamics Analysis

The adsorption thermodynamics of the carbonized ZIF−8@PAN nanofiber membrane for Cd^2+^ was analyzed by the Gibbs free energy formula at three temperatures of 20 °C (293 K), 25 °C (298 K), and 35 °C (308 K), respectively. The obtained thermodynamic data is plotted and fitted in Figure 8. Some adsorption thermodynamic parameters are listed in Table 1.

It can be seen in Table 1 that all of the calculated Δ*G^θ^* values at various temperatures were negative, indicating that the Cd^2+^ adsorption process of the carbonized ZIF−8@PAN nanofiber membrane was spontaneous and feasible, which theoretically proves the adsorption ability of the carbonized ZIF−8@PAN nanofiber membrane for Cd^2+^ in aqueous solution. Δ*H**^θ^* was positive, as seen in Table 1, indicating that the adsorption of the carbonized ZIF−8@PAN nanofiber membrane for Cd^2+^ is an endothermic process and the increasing temperature is conducive to Cd^2+^ adsorption. It is suggested that the mass transfer resistance is produced when Cd^2+^ displaces from the aqueous solution to the surface of the carbonized ZIF−8@PAN nanofiber membrane. Therefore, the higher energy is required to overcome the mass transfer resistance besides for the energy released by the combination of Cd^2+^ and the adsorbent surface. Additionally, Δ*S**^θ^* was also positive, indicating that the disorder degree at the solid–liquid two phase interface increases during the adsorption process [46].

### 3.5. Effect of pH

The effect of pH on the Cd^2+^ adsorption performance of the carbonized ZIF−8@PAN nanofiber membrane is shown in Figure 9. Under acidic condition (pH = 5.5) at the adsorption temperature of 35 °C, the Cd^2+^ concentration changed very little, as seen in Figure 9a. At this time, the *Q_e_* value and adsorption efficiency were zero, respectively, as shown in Figure 9b and c. The reason is that both free H^+^ in acidic solution and Cd^2+^ are cationic, and thus both of them could be absorbed by the carbonized ZIF−8@PAN nanofiber membrane and compete with each other during the adsorption process. Compared with Cd^2+^, H^+^ was easier to be adsorbed by our adsorbent because it has a smaller radius than that of Cd^2+^, which gives it faster transfer speed from the standard solution to the surface of our adsorbent. In Figure 9b,c, with increasing pH, the *Q_e_* value was about 102 mg L^−1^, and the Cd^2+^ adsorption efficiency also increased and could reach about 90% at pH = 7.5. However, as the pH continued to increase to 8.5, both the *Q**_e_* value and the adsorption efficiency decreased, while the Cd^2+^ adsorption equilibrium time was also shortened to no more than 10 h. The reason is that in an alkaline solution, compared with our adsorbent, Cd^2+^ was easier to interact with free OH^–^ and generated white precipitation of Cd(OH)_2_, which leads to a remarkable decrease in the initial Cd^2+^ concentration.

### 3.6. Adsorption Mechanism

In order to probe the adsorption mechanism of the carbonized ZIF−8@PAN nanofiber membrane for Cd^2+^, a Cd^2+^ adsorption experiment was carried out under the optimal condition of pH value of 7.5 and the adsorption temperature of 35 °C. Figure 10 shows the change of Cd^2+^ adsorption concentration, *Q_e_*, and the adsorption efficiency of the carbonized ZIF−8@PAN nanofiber membrane with the increasing of adsorption time under this optimal condition. As shown in Figure 10a, in the first 7 h, the Cd^2+^ concentration rapidly declined, while the Cd^2+^ adsorption rate slowed down after 7 h. This may be because the initial Cd^2+^ concentration was relatively high and could fully contact the adsorbent surface. With the increasing of adsorption time, a lot of Cd^2+^ entered into the internal pores of the adsorbent and occupied more adsorption active points, which causes adsorption to slow. In Figure 10b,c, it can be observed that the *Q_e_* value was 102 mg L^−1^, and the absorption efficiency was more than 90% when the adsorption equilibrium is reached under the optimal adsorption condition.

According to the data from Figure 10, the Cd^2+^ adsorption process was fitted with the pseudo first-order and second-order adsorption equations, respectively.

Figure 11 shows the linear fitting curves of the pseudo first-order and second-order adsorption kinetic models of the carbonized ZIF−8@PAN nanofiber membrane. In Figure 11, R^2^ is the relatively linear coefficient, indicating the deviation degree of the experimental data from the fitting curve of the adsorption kinetic equation. Compared with the R^2^ value (0.99566) in Figure 11a, the R^2^ value (0.99956) in Figure 11b is closer to 1, meaning that the pseudo second-order adsorption is more suitable for describing the Cd^2+^ adsorption process of the carbonized ZIF−8@PAN nanofiber membrane. This suggests that the Cd^2+^ adsorption process of our adsorbent was accompanied by chemical adsorption in addition to physical adsorption [40].

## 4. Conclusions

In this paper, a ZIF−8@PAN nanofiber membrane was fabricated via in situ solvothermal reaction. During the fabrication process, the metal ion source of ZIF−8 was introduced on the surface of the PAN nanofiber membrane by means of the environmentally friendly magnetron sputtering technology. Then, the ZIF−8@PAN nanofiber membrane was calcinated in a tube furnace at 900 °C to achieve excellent Cd^2+^ adsorption performance. The results showed that polyhedral ZIF−8 could uniformly be fixed onto the surface of the PAN electrospun nanofiber membrane. After carbonization, the profile of ZIF−8 on the nanofiber membrane surface shrank. Compared with the nanofiber membrane without being carbonized, the carbonized ZIF−8@PAN nanofiber membrane had higher Cd^2+^ adsorption ability. Under the optimal adsorption condition of the adsorption temperature of 35 °C and pH of 7.5, the *Q_e_* value was about 102 mg L^−1^, and the Cd^2+^ adsorption efficiency was more than 90% after the adsorption equilibrium was reached. On the basis of the adsorption thermodynamics analysis, the Cd^2+^ adsorption process of the carbonized ZIF−8@PAN nanofiber membrane was spontaneous and feasible. The Cd^2+^ adsorption process of our carbonized adsorbent was more in line with the second-order adsorption kinetic model, which indicates that there exists chemical adsorption besides for physical adsorption during the Cd^2+^ adsorption. It is hoped that this novel carbonized nanofiber membrane may be potentially developed into a new type of material for the adsorption of heavy metal Cd^2+^ pollutants from wastewater.

## Figures and Tables

**Figure 1 polymers-14-02523-f001:**
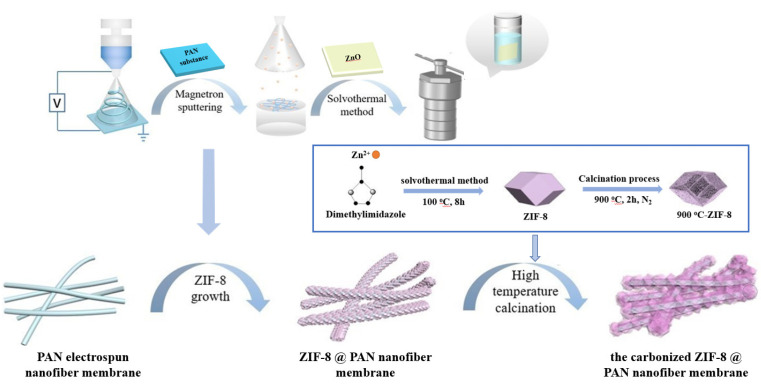
Fabrication route of the carbonized ZIF−8@PAN nanofiber membrane.

**Figure 2 polymers-14-02523-f002:**
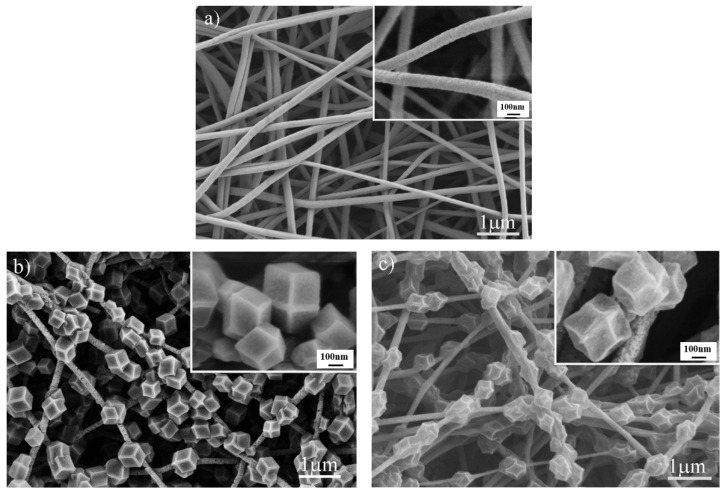
SEM images of original PAN electrospun nanofiber membrane (**a**), the ZIF−8@PAN nanofiber membrane (**b**), and the carbonized ZIF−8@PAN nanofiber membrane (**c**).

**Figure 3 polymers-14-02523-f003:**
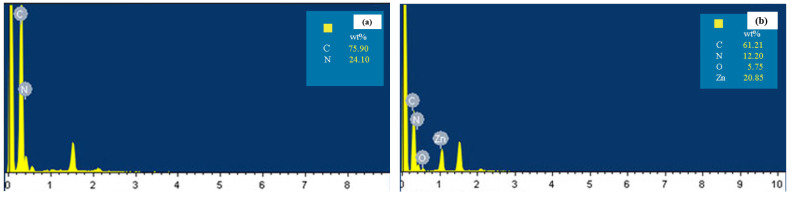
EDS spectra of the original PAN electrospun nanofiber membrane (**a**), ZIF−8@PAN nanofiber membrane (**b**), and the carbonized ZIF−8@PAN nanofiber membrane (**c**).

**Figure 4 polymers-14-02523-f004:**
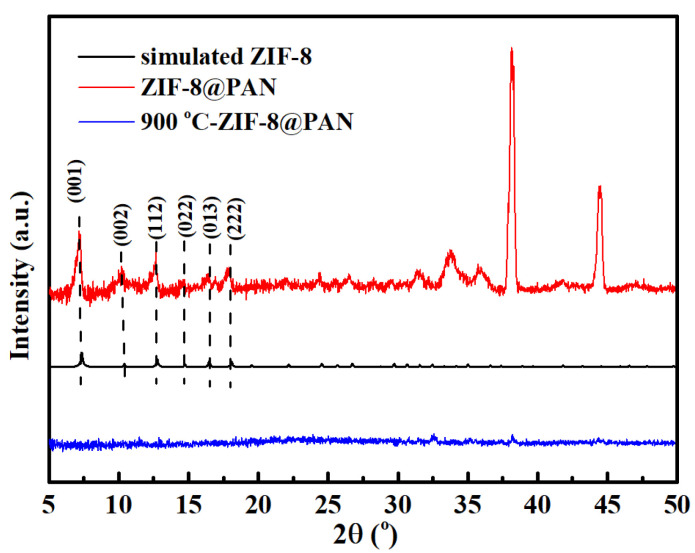
XRD patterns of the ZIF−8@PAN nanofiber membrane, the simulated ZIF−−8, and the carbonized ZIF−8@PAN nanofiber membrane.

**Figure 5 polymers-14-02523-f005:**
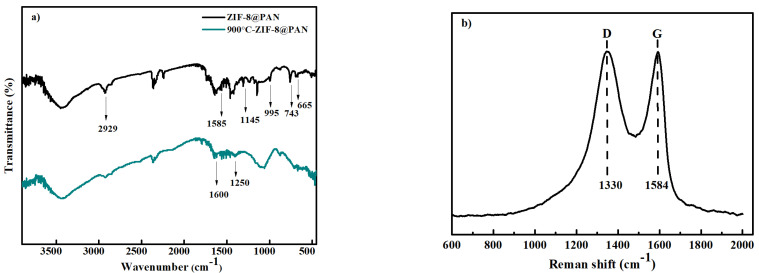
(**a**) FTIR spectra of ZIF−8@PAN nanofiber membrane and the carbonized ZIF−8@PAN nanofiber membrane, and (**b**) Raman spectrum of the carbonized ZIF−8@PAN nanofiber membrane.

**Figure 6 polymers-14-02523-f006:**
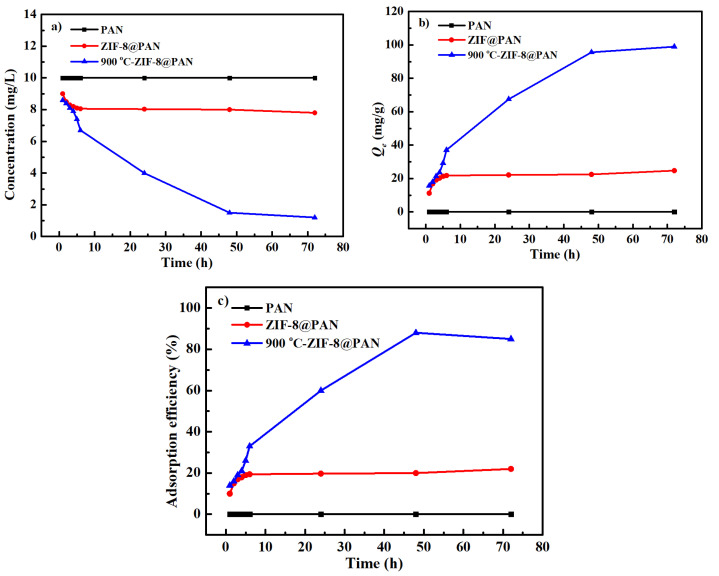
The Cd^2+^ adsorption concentration (**a**), adsorption capacity *Q_e_* (**b**), and adsorption efficiency (**c**) of original PAN nanofiber membrane, ZIF−8@PAN nanofiber membrane, and the carbonized ZIF−8@PAN nanofiber membrane.

**Figure 7 polymers-14-02523-f007:**
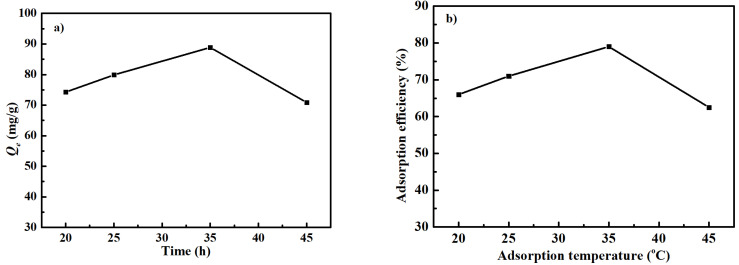
The Cd^2+^ adsorption capacity *Q_e_* (**a**) and adsorption efficiency (**b**) of the carbonized ZIF−8@PAN nanofiber membrane at various adsorption temperatures and pH = 6.5.

**Figure 8 polymers-14-02523-f008:**
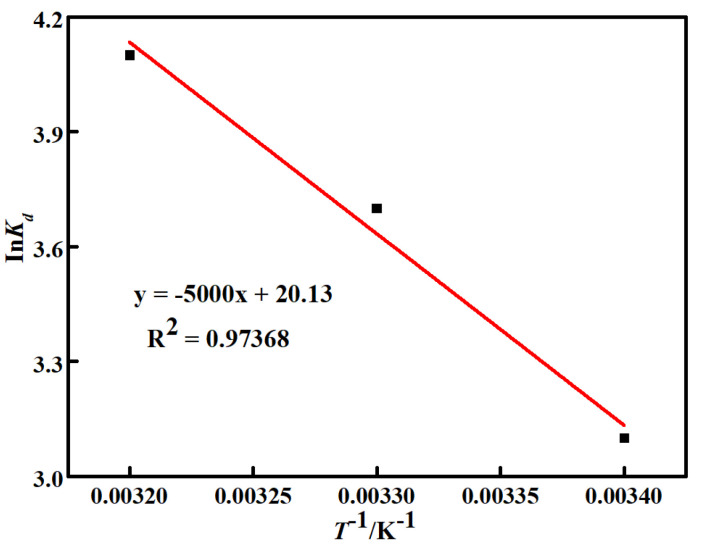
The fitted Cd^2+^ adsorption thermodynamic curve of the carbonized ZIF−8@PAN nanofiber membrane at different temperatures.

**Figure 9 polymers-14-02523-f009:**
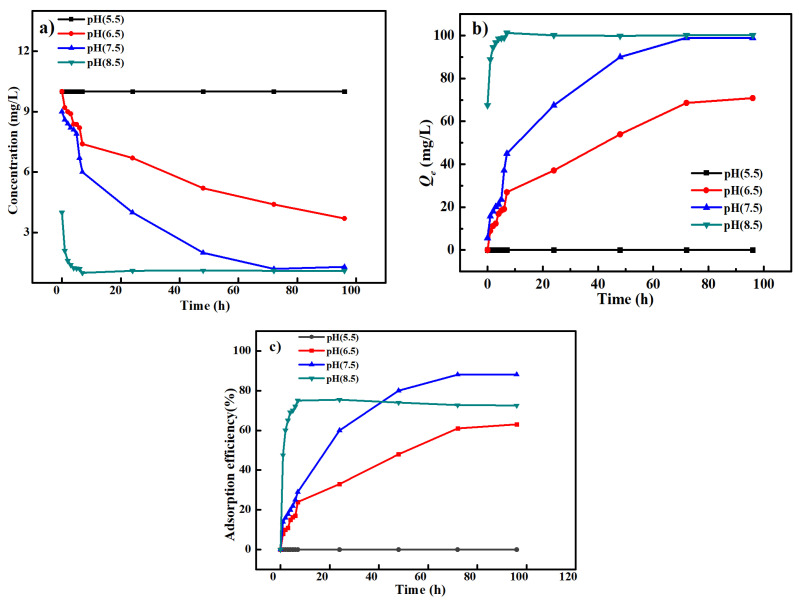
The Cd^2+^ adsorption concentration (**a**), adsorption capacity *Q_e_* (**b**), and adsorption efficiency (**c**) of the carbonized ZIF−8@PAN nanofiber membrane at different pH values and the adsorption temperature of 35 °C.

**Figure 10 polymers-14-02523-f010:**
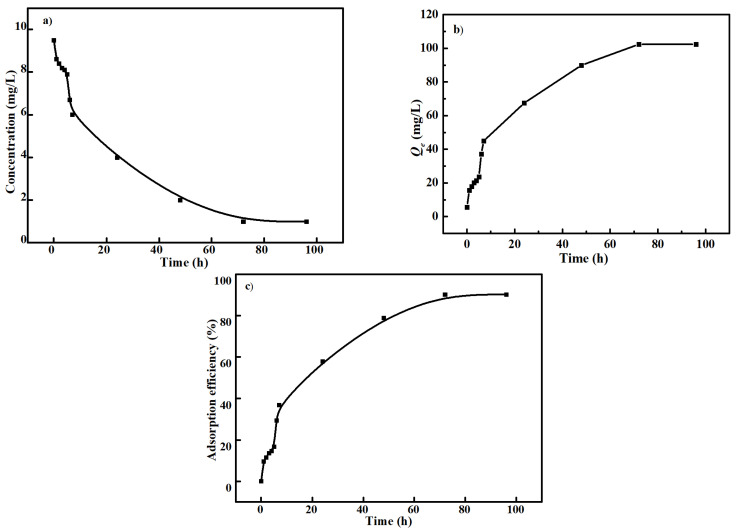
The Cd^2+^ adsorption concentration (**a**), adsorption capacity *Q_e_* (**b**), and adsorption efficiency (**c**) of the carbonized ZIF−8@PAN nanofiber membrane under the conditions of pH of 7.5 and water bath temperature of 35 °C.

**Figure 11 polymers-14-02523-f011:**
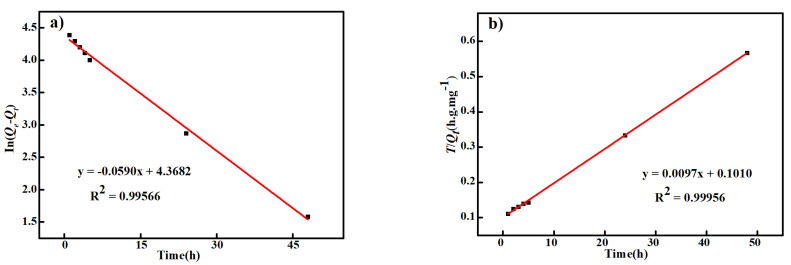
Linear fitting curves of pseudo first-order (**a**) and second-order (**b**) adsorption kinetic models of the carbonized ZIF−8@PAN nanofiber membrane.

**Table 1 polymers-14-02523-t001:** The Cd^2+^ adsorption thermodynamic parameters of the carbonized ZIF−8@PAN nanofiber membrane at different temperatures.

Δ*H^θ^*/(kJ·mol^−1^)	Δ*S**^θ^*/(J·mol^−1^·K^−1^)	Δ*G^θ^*/(kJ·mol^−1^)		
		293 K	298 K	308 K
41.6	167.1	−7.6	−9.2	−10.5

## Data Availability

The data presented in this study are available upon request from the corresponding author.

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
