# Peer review of "Preparation of the Carbonized Zif−8@PAN Nanofiber Membrane for Cadmium Ion Adsorption"

_polymers, 2022, doi:10.3390/polym14132523_

Round 1
Reviewer 1 Report
Comments on the “Preparation of Zif-8 @ PAN Porous Carbon Nanofiber Membrane for Cadmium Ion Adsorption”. In this paper, the zeolitic imidazolate framework (ZIF-8) @ polyacrylonitrile (PAN) nanofiber membrane was prepared and carbonized for heavy metal cadmium ion (Cd2+) adsorption in aqueous medium. While some major revisions should be made before being further processed. The specific comments are list as follows:
1. The electrospinning method has been widely used for fabricating fibrous membranes that could be used in various aspects including but not limited to separation, air filtration, drug delivery system, tissue engineering, antimicrobial applications, catalysis and energy storage devices. Papers related to the fibrous membranes with controllable water absorption capacity are recommended to enhance the literature research in the introduction section. ( DOI: 10.1080/15435075.2021.1941046; https://doi.org/10.1016/j.jpowsour.2022.231121; https://doi.org/10.1016/j.ijhydene.2021.05.121; https://doi.org/10.1016/j.ijheatmasstransfer.2020.119853; https://doi.org/10.1016/j.matlet.2020.127759. )
2. Some format mistake should be carefully checked before submission, such as the first subtitle in section 3 should be 3.1 rather than 2.7.
3. It is stated in section 3 that “After the solvothermal reaction, as shown in Figure 2(b), a layer of ZIF-8 polyhedral particles with uniform size are attached onto the surface of the PAN nanofiber membrane. The enlarged image in the upper right corner on Figure 2(b) clearly shows that the particles have a rhombic dodecahedron structure, which is consistent with the reported references [1-2] and confirms that ZIF-8 is successfully synthesized and fixed on the surface of the PAN nanofiber membrane via in situ growth.” Generally, when confirming the component of a material, the elemental analysis or phase characterization should be conducted. It is not rigorous to make a conclusion without enough evidence. My suggestion is revising the description in the SEM characterization with some assumptions and then prove in the EDS and XRD parts.
4. “EDS was performed to investigate the element species of nanofiber membrane before 196 and after functional modification, and the results were shown in Figure 3.” When first introducing the EDS, the full name should be displayed.
5. When displaying the XRD patterns, the index information should be offered in the diagram or the corresponding descriptions.
6. The manuscript should be carefully checked before submission, the content in section 3.2 seems to be incomplete and the description of Fig.6 should be placed in front of the diagram.
7. The content that describing the Fig.6b is missing, double check is necessary before final submission.
8. No paper published after 2020 was cited in this work, which makes the timeless of this research work very weak. More recent published research papers should be cited to enhance the timeless of this work.
Reviewer 2 Report
The manuscript has potential, but the approach is very impressive. In the introduction the authors report similar materials with adsorption capacities above 200 mg/g. Despite the appropriate approach in the introduction, the authors do not present the values ​​in adsorption capacity (mg/g) throughout the manuscript. Values ​​expressed only in removal efficiency (%), do not present the necessary precision for an original manuscript. Please correct the entire document.
Characterizations that confirm the information that the authors comment on in the introduction must be made. For example, high surface area and thermal stability.
In general, the equations of the mathematical models should appear in the methodology and not in the discussion of the results.
Figure 6 should be rewritten so that it is qt (adsorption capacity at time t) on the Y axis by time on the X axis.
Normally, the parameters Qe and Ce of the adsorption equilibrium isotherm analysis are used to make the thermodynamic adjustments. Please rewrite the text so that it is clear how the adjustments were made.
The processing of the adsorbent material reported in this work is a little complex, please compare with similar materials and discuss the advantages of using this material and not a simpler material, as presented in several other articles.
Reviewer 3 Report
The submitted manuscript for review, in my opinion, needs to be improved. So most of the drawings need to be improved (increase the font size, line thickness, etc.). Further, I recommend the authors to deepen the explanation of the results obtained and support them with literary sources. I lacked the data of mechanical studies in the presented work. As well as micrographs proving the porous morphology of the resulting membranes. Below is a list of questions and comments that need to be addressed: Lines 14, 15. "Then, ZIF-8@PAN nanofiber membrane was prepared via the in situ solvothermal reaction and carbonized in a tube furnace at 900 °C under a N2" Hence, I recommend the authors to remove the name of the polymer - PAN from the title of the article. Keywords: "PAN nanofiber membrane" - remove Lines 38-40. Please add a link. 2.1. materials. It is necessary to add information about the polymer and the production of PAN nanofibers. Figure 1. The figure is not informative. I recommend increasing the font size. Lines 142-144. it is desirable to indicate the method of shooting. Figure 2. Specify the scale for photos inserted in the upper right corner of the SEM photos. Line 195. It is not clear from these micrographs that the resulting fibers are porous. Therefore, I propose here and elsewhere to omit the definition - "porous" or use a different term or designation. Figure 3. Need to improve the quality of the drawings. Lines 209-221, Figure 4. In my opinion, the description of the obtained diffractograms needs to be redone and described in more detail. At present, shutters focus only on the angular position of the peaks. In the presented diffraction pattern for carbonized material, a broad peak is observed, which characterizes the amorphous carbon phase (which is described in the work - https://doi.org/10.3390/polym13040537). For graphite-like structures, the diffractogram will have a different form, for example, as here - https://doi.org/10.1088/1755-1315/316/1/012032 I recommend that the authors familiarize themselves with the examples given and change the description of the obtained diffractograms. Also, the authors should take into account that the processing temperature of the precursor material is insufficient to obtain graphitized materials. Also, it may make sense to start the description of the results of the work with the SAR data. Which speak of the amorphous nature of the resulting material. Further we can develop the theme of porous carbon material... Lines 233-235. "Meanwhile, two new characteristic peaks appear at 1560 cm-1 and 1260 cm-1, which is explained by the formation of 234 a porous carbon structure after high temperature calcination of ZIF-8@PAN nanofiber membrane." - It is not entirely clear how the peaks explain the material's porosity. Authors can provide references or explanation. Figure 6. Need to improve the quality of the drawings. Figures 9-11. The quality of the drawings needs to be improved. Lines 409, 410. Vera, B.; Mikhail, S.; Alexander, G.; Kirill, L.; Carlo, L. Metal-organic frameworks: structure, properties, methods of synthesis and characterization. Russ. Chem. Rev. 2016, 85, 280-307. - Names and surnames are mixed up in the link. Names are given in full, surnames are abbreviated.
Round 2
Reviewer 1 Report
This version has been quite improved after revision and some format issue should be careful, such as the style of the last two reference. Especially, the 47th refernce is missing. And some references are marked with underline. Another is the XRD data, the pdf card information should be offered to prove that the indexed crystal plane in the XRD patterns could be traced. the last is the Raman spectrum, the specific position of D and G peak should be offered to confirm wether some shift happens.
Reviewer 2 Report
The authors made some important changes to the manuscript. However, the main gap in this work is not considering the adsorption capacity in the evaluation of the adsorbent potential.
Removal efficiency demonstrates that the material is capable of retaining a certain percentage of particles until it reaches equilibrium. This is a very important parameter. Even so, the following analysis must be performed by the authors: the value of 99% removal of a given pollutant can be obtained with 1g of adsorbent "X" and with 1kg of adsorbent "Y", the real potential of the adsorbent must be supplemented with the value in mg of adsorbate/g of adsorbent.
The adsorption capacity presents values in mg/g, that is, the value in mass of adsorbed particles by the value in mass of adsorbent. Without the graphs and complete analysis of this basic result, I do not recommend this manuscript for publication.
Reviewer 3 Report
After the first review, the authors made some changes to the manuscript, but at the moment a number of questions remain open and require an answer to them before the publication of the manuscript. Lines 38-40. What porosity are the authors talking about? In nanofibers or the distance between them? The authors did not support this statement with a link! Line 121. The authors did not indicate whether they used a homopolymer or a copolymer (its composition). The authors need to indicate what concentration the spinning solution had and how the solvent was removed! This is important as it determines whether the morphology will be porous or not... Line 135. "a in suit" needs to be corrected Lines 215-217. "Figure 2 shows SEM images of the original PAN, ZIF-8@PAN and ZIF-8@PAN porous carbon nanofiber membranes, respectively. In Figure 2(a), the surface of PAN nanofiber membrane is relatively smooth" - the authors themselves contradict talking about porous fibers and that their surface is smooth. Also, in my opinion, it is not correct to introduce the term porous until it is shown or proven in the present work. Figure 4. The new diffractograms presented show no carbon structure at all. I do not understand such a diffraction pattern and ask the authors to explain it in detail. Line 275. It is necessary to remove the extra point. Lines 279, 280. "Raman spectrum of ZIF-8@PAN porous carbon nanofiber membrane was carried out to confirm the formation of graphitized structure." - the authors have previously operated with the term carbonization, but here they already talk about graphitization. Again, I do not agree that a graphite-like structure formed at 900 ° C, otherwise this fact must be proved. The authors should take into account the elemental composition, which they show earlier in Figure 3. It is also difficult for me to correlate the above paragraph with the data of X-ray diffraction analysis. The type of curves in Figs. 9 and 10 should be the same?!Author Response
Please see the attachment.

Round 3
Reviewer 2 Report
The authors made the necessary improvements to increase the potential of the article. I now recommend this manuscript for publication.
Reviewer 3 Report
The authors took into account previous comments, a number of additional suggestions are given below: Line 2. "NanofiberMembrane" - you need to add a space Line 26. "electrospinning nanofiber membrane" suggest replacing with "Electrospun nanofiber membrane" Line 260. You need to add a link, an example was already given to the authors earlier (https://doi.org/10.3390/polym13040537). Lines 264, 266. "amorphous graphitized structure" - "ZIF-8 has been carbonized" - I recommend paraphrasing.